## [Transparent Peer Review file · Nature Communications]

Prussian Blue Nanoparticles Targeting Multiple PANoptosome-Mediated PANoptosis for Myocardial Ischemia-Reperfusion Injury Therapy

Corresponding Author: Professor Yuanyi Zheng

Version 0:

Reviewer comments:

Reviewer #1

(Remarks to the Author)

Xu et al. devised an innovative method for synthesizing Prussian blue (PB) nanoparticles, which effectively inhibit PANoptosis, thereby comprehensively blocking pyroptosis, apoptosis, and necroptosis and preventing their crosstalk in the treatment of myocardial ischemia-reperfusion injury (MIRI). The concept is OK, some revision should be done to improve the quality of this manuscript.

1. The authors' assertion that myocardial ischemia-reperfusion injury (MIRI) is closely related to PANoptosis by single-cell sequencing alone lacks sufficient evidence. The genes involved in PANoptosis are similar to those involved in pyroptosis, apoptosis, and necrotic apoptosis, which makes it difficult to distinguish between the different forms of cell death. Furthermore, the hallmark of PANoptosis is the assembly and activation of the PANoptosome, which has yet to be directly characterised. The study lacks direct characterization of the PANoptosome.

2. In light of the assumption that PANoptosis can be triggered by MIRI, it is imperative for the authors to elucidate the mechanism by which PANoptosis is initiated in the context of MIRI. This is a pivotal step in identifying the specific PANoptosis pathway that is activated, given that the modes of activation and triggering mechanisms of different PANoptosomes vary.

3. It is not sufficient to conclude that PB can inhibit the formation of multiple PANoptosomes based on the evidence that PB reacts with AIM2, ZBP1, and RIPK1. The core components of the four PANoptosomes are highly similar, with AIM2, ZBP1 and RIPK1 representing the core components of the AIM2-PANoptosome. Therefore, it is unclear how the authors can claim to inhibit multiple PANoptosomes simultaneously. Once more, we propose that the authors elucidate the specific type of PANoptosis precipitated by MIRI.

4. The article's primary focus is on sequencing and molecular modelling, which illustrate the inhibitory effect of PB on PANoptosis, which seriously lacks experimental validation and molecular mechanism. It would be advisable to confirm the inhibitory effect of PB on the assembly of the PANoptosome directly, for example through fluorescence co-localization or immunoprecipitation experiments.

5. Please explain the rationale behind selecting the third day after surgery as the time point for tissue protein detection. Does the expression of PANoptosis-related proteins exhibit a greater degree of significance at either a shorter or longer time point?

6. MIRI is a common occurrence in cardiomyocytes situated within the coronary artery reperfusion zone. How do the authors distinguish these cells in vivo in order to conduct the subsequent step of the experiment?

7. The authors conducted fluorescence imaging of the principal organs of MIRI mice at 12 and 24 hours post-injection of Cy5.5-labeled PB@PM. Nevertheless, the findings indicated that only a minimal quantity of PB@PM remained within the heart at the 24-hour mark. It would be valuable to ascertain whether any remaining PB@PM is present in the heart after 28 days. Given that Cy5.5-labelled PB is not enriched in the heart, it would be reasonable to question its role in cardiac remodeling. Furthermore, it would be beneficial to determine whether the enrichment of PB@ in the liver causes any toxic side effects to the liver.

8. As illustrated in Figure 5, PB has been demonstrated to enhance cardiac function and ventricular remodeling, whilst concurrently reducing immune and inflammatory responses. However, it remains unclear why PB does not demonstrate a robust anti-PANoptosis effect in vivo. What is the precise function of PB in enhancing cardiac performance and preventing ventricular remodeling? Please elucidate the advantages and necessity of utilizing PB@PM.

9. The authors applied RNA sequencing and protein profiling to ischemic heart tissue in order to demonstrate the inhibitory

effects of PANoptosis on PB@PM. It would be beneficial to supplement the immunohistochemistry of representative pan-apoptotic proteins in heart tissue, as well as to ascertain whether downstream inflammatory factors will be affected.

Reviewer #2

(Remarks to the Author)

Comment: The manuscript entitled "Nanoparticles Targeting Multiple PANoptosome-Mediated PANoptosis for Myocardial Ischemia-Reperfusion Injury Therapy" is novel and creative. The results found that PB@PM as an effective PANoptosis inhibitor and revealing its mechanism of action in MIRI therapy. I think this article can be accepted after minor revisions. Some suggested modifications are as follows:

1. The title should indicate that this nanomaterial is PB@PM;
2. During in vivo experiments, PB and PB@PM. What is the concentration determined based on?
3. Supplementary image: In all bar charts, the horizontal line representing the difference between the two groups needs to be further away from the error bar
4. Please supplement the contributions of other authors
5. Please check the format of the full-text references, such as reference 22, Qi Z, Zhu L, Wang K, Wang N. PANoptosis: Emerging mechanisms and disease implications. *Life Sci.* 2023;333:122158.
6. The section of results should be supplemented with ns: $P > 0.05$
7. In Fig. S11, IL-10 and TGF- β are both M2 macrophage markers and can promote extracellular matrix repair, supplementing TGF- β data.

Reviewer #3

(Remarks to the Author)

This study by Xu et. al innovatively proposes a PANoptosis inhibition strategy based on nano-Prussian blue (PB) to address the clinical challenge of myocardial ischemia-reperfusion injury (MIRI). Firstly, the authors defined MIRI as a PANoptosis-related disease using single-cell sequencing analysis. Subsequently, the research team used molecular dynamics simulations to confirm the binding ability of PB to PANoptosis key proteins RIPK1, ZBP1 and AIM2, thereby comprehensively inhibiting PANoptosis. On this basis, the authors further constructed platelet membrane-modified PB nanoparticles (PB@PM), which successfully realized the targeting of myocardial injury zone. As a PANoptosis inhibitor, PB@PM significantly improved cardiac function in the MIRI mouse model and promoted extracellular matrix repair and neovascularization. The mechanism of PB@PM in inhibiting PANoptosis and the crosstalk between pyroptosis, apoptosis, and necroptosis was comprehensively revealed through single-cell sequencing, molecular biology techniques, and transcriptome sequencing analysis. Overall, this is a very interesting and well-researched study that can appeal to readers from a wide range of fields including cardiovascular, nanomedicine and biomaterials. The experimental design, description, and implementation are reasonable, and the overall interpretation is also reasonable. This study may provide a framework for studying nanobiological interactions to uncover broader nanomaterial mechanisms. I think this study is suitable for publication in *Nature Communications*, after addressing the following questions:

1. Molecular dynamics simulations demonstrated the binding of PB to the target protein. This is the result of the simulation. The author should provide experimental data on PB binding to target proteins, such as surface plasmon resonance.
2. The authors should add the safety of the nanomedicine, such as pathology tests and blood indicators.
3. In Figure 5I, the image is too small and it is recommended to highlight the key signals.
4. In cellular experiments, in addition to the three key proteins (RIPK1, ZBP1 and AIM2), the authors also needed to select some key downstream pyroptosis, apoptosis, and necroptosis-related proteins for validation.
5. Some spelling errors need to be corrected. It looks like some of the uploaded videos are named incorrectly (ZBP1 and PB, RIPK1 and PB) and need to be corrected. "RIPK1 PANoptosome" is incorrectly written in Figure 1.
6. In ex vivo and in vivo therapeutic experiments, what was the main basis for the authors in defining treatment time points and concentrations?
7. The XPS data requires peak splitting processing.
8. The authors should appropriately describe the remaining limitations of the study.

Version 1:

Reviewer comments:

Reviewer #3

(Remarks to the Author)

The manuscript has undergone significant improvements and enhancements, and I have no issues. Thank you to the authors for their efforts.

Reviewer #4

(Remarks to the Author)

This study systematically investigated the therapeutic mechanism of Prussian blue nanoparticles (PB@PM) targeting PANoptosis in myocardial ischemia-reperfusion injury (MIRI). Through analyzing human cardiac scRNA-seq data, significant upregulation of PANoptosis-related genes in MIRI was confirmed, followed by molecular dynamics simulations to

predict specific binding between PB and PANoptosome core proteins. Further, the authors demonstrated the efficacy of platelet membrane-coated nanoparticles (PB@PM) in suppressing PANoptosis activation and assembly in vitro and in vivo. Accordingly, the authors then hypothesized that PB@PM concurrently targets RIPK1/ZBP1/AIM2 within the PANoptosome, disrupting crosstalk among pyroptosis, apoptosis, and necroptosis to confer cardioprotection. However, the claim that PB@PM "blocks PANoptosome assembly" lacks direct experimental validation. Besides, this study did not explore whether PB@PM reprograms mitochondrial metabolism or inhibits mtDNA release, which are key triggers of PANoptosis. Meanwhile, merely downstream moleculars related to PANoptosis were observed while the upstream mechanism regulating PANoptosis was not elucidated. Generally speaking, the innovation of this study was average and the experimental design was not persuasive enough.

Question 1: The image quality needs improvement. Please add scale bars to the ultrasound images (Fig 5) and check the scale annotations in other figures.

Question 2: Both the levels of apoptotic caspases and their cleaved active fragments are low under normal conditions, while the accumulation of cleaved fragments post myocardial infarction might reduce the total protein levels. Please verify the Western blot (WB) data and ensure that both full-length and cleaved fragments are displayed on the same membrane.

Question 3: In Fig 3, all the molecular dynamics simulations for Prussian blue (PB) binding to PANoptosis-related molecules used a 200-ns timeframe. However, the RMSD of the PB-RIPK1 complex did not reach equilibrium by 200 ns. Please reevaluate whether 200 ns is suitable to the PB-RIPK1 system.

Question 4: The efficiency of PANoptosis inhibition by PB@PM was similar to the other three specific programmed cell death inhibitors. What is the advantage of using PB@PM over these inhibitors? Please clarify the unique therapeutic value of PB@PM in PANoptosis modulation.

Question 5: How long does PB@PM persist in cardiac tissue? Additionally, please explain the primary mechanism by which PB@PM ameliorates fibrosis at the 28-day timepoint (e.g., modulation of inflammatory pathways, suppression of PANoptosis-related cytokines, or direct effects on fibroblast activation)

Question 6: Mitochondrial damage plays a central role in PANoptosis. The authors claimed that PB@PM alleviates mitochondrial damage and inflammation, but did not explore the core mechanism of PB@PM ameliorating PANoptosis. Does PB@PM directly target PANoptosis or act through regulating mitochondrial dysfunction to suppress PANoptosis?

Question 7: What is the underlying mechanism of PB@PM reprogramming mitochondria metabolism? Does it directly bind with mitochondrial proteins or modulate mtDNA release?

Question 8: Whether PB@PM truly block PANoptosome assembly or not should be further verified. Current data only showed inhibition of downstream cell death markers, but failed to demonstrate the disruption of PANoptosome complex.

Question 9: Molecular dynamics simulations suggested that PB binds with RIPK1, ZBP1 and AIM2, while the direct evidence of PB disrupting PANoptosome complex assembly was absent.

Question 10: The observed reduction in TNF- α /IL-1 β by PB@PM derives from PANoptosis inhibition or direct immune suppression should be further elucidated.

Version 2:

Reviewer comments:

Reviewer #3

(Remarks to the Author)

The authors have taken the reviewer's rigorous critique seriously and have performed substantial additional work to elevate the manuscript. Based on a comprehensive review of the author's point-by-point rebuttal to Reviewer #4's comments, the revised main manuscript, and the supplementary information, this revised submission demonstrates a significant and satisfactory improvement in addressing the reviewer's major concerns. The overall quality, mechanistic depth, and evidence supporting the central claims have been substantially strengthened. The revisions have transformed a study with potential into one with solid, multi-faceted evidence for its central thesis: that PB@PM is a multi-target nano-therapeutic that mitigates MIRI by disrupting PANoptosis through both direct molecular interference and mitochondrial protection.

The study's novelty, integrative methodology, therapeutic relevance, and significantly improved mechanistic depth now align with the high standards of Nature Communications. The work provides valuable insights into PANoptosis in cardiovascular disease and presents a promising nanomedicine strategy.

Recommendation: Accept for publication in Nature Communications.

Comprehensive point-by-point response to reviewers' comments

Reviewer #1 (Remarks to the Author):

Xu et al. devised an innovative method for synthesizing Prussian blue (PB) nanoparticles, which effectively inhibit PANoptosis, thereby comprehensively blocking pyroptosis, apoptosis, and necroptosis and preventing their crosstalk in the treatment of myocardial ischemia-reperfusion injury (MIRI). The concept is OK, some revision should be done to improve the quality of this manuscript.

Response: Dear reviewer, we express our sincere gratitude for your meticulous evaluation and acknowledge your recognition of our investigative efforts. We have rigorously evaluated your valuable suggestions and comments and have implemented substantive revisions throughout the manuscript.

Comment 1. The authors' assertion that myocardial ischemia-reperfusion injury (MIRI) is closely related to PANoptosis by single-cell sequencing alone lacks sufficient evidence. The genes involved in PANoptosis are similar to those involved in pyroptosis, apoptosis, and necrotic apoptosis, which makes it difficult to distinguish between the different forms of cell death. Furthermore, the hallmark of PANoptosis is the assembly and activation of the PANoptosome, which has yet to be directly characterised. The study lacks direct characterization of the PANoptosome.

Response: We appreciate the reviewer's insightful comments. While PANoptosis integrates molecular components of pyroptosis, apoptosis, and necroptosis, its unique hallmark lies in the assembly of PANoptosomes, which orchestrate crosstalk among these pathways. To substantiate the link between MIRI and PANoptosis beyond single-cell sequencing, we performed the following experiments:

(1) Human Heart Single-Nuclear Transcriptomics: Analysis of human heart samples revealed significant upregulation of PANoptosis-related genes (e.g., *AIM2*, *ZBP1*, *RIPK1*, *NLRP3*, caspases 1/3/8, *GSDMD*, *MLKL*) in cardiomyocytes of the ischemic zone (IZ), with apoptosis being the dominant form (**Fig. 2F, S3–S5**). Subcluster analysis

further identified cardiomyocyte subpopulations (e.g., clusters 1, 2, 4) with elevated PANoptosis activity, correlating with cell loss post-ischemia (**Fig. 2G–K**).

(2) In Vivo Validation in MIRI Mice: Transcriptomic and proteomic analyses demonstrated that MIRI upregulated PANoptosome components, including AIM2, ZBP1, RIPK1, NLRP3, ASC, FADD, cleaved caspases (1/3/8), and phosphorylated RIPK3/MLKL (**Fig. 6J, S18–S19**). PB@PM treatment significantly suppressed these markers, confirming PANoptosome activation in MIRI and its inhibition by PB@PM.

(3) In Vitro Co-Localization and Functional Assays: Fluorescence imaging confirmed PB's co-localization with AIM2, ZBP1, and phosphorylated RIPK1 in cardiomyocytes (**Fig. S20A**). Overexpression of AIM2/ZBP1/RIPK1 in HL-1 cells showed that PB counteracted OGD/R-induced upregulation of these proteins and their phosphorylated forms (**Fig. S20B**). Additionally, PB@PM rescued OGD/R-induced mitochondrial dysfunction and reduced apoptosis, pyroptotic membrane pores, and necroptotic markers (e.g., p-MLKL) (**Fig. S21**).

(3) Combination Therapy Validation: Co-treatment with inhibitors of pyroptosis (Disulfiram), apoptosis (Z-VAD-FMK), and necroptosis (Necrostatin-1) mimicked PB@PM's therapeutic effect, highlighting PANoptosis crosstalk in MIRI (**Fig. S21D**).

These findings collectively demonstrate that MIRI activates multiple PANoptosomes (AIM2, ZBP1, RIPK1) and that PB@PM broadly inhibits their assembly and downstream death pathways.

Fig.S18 PB@PM inhibits PANoptosis-related protein activation. Semi-quantitative analysis of AIM2, ZBP1, Pyrin, ASC, FADD, Bax, and Bcl-2, phosphorylated/total RIPK1/RIPK3/MLKL, and cleaved/total caspase 1/3/8 and GSDMD in ischemic myocardium on 3 days post-I/R or sham surgery ($n = 5$ biologically independent replicates). Data: mean \pm SD. Significance: one-way ANOVA with Tukey's post-hoc test (* $P < 0.05$, ** $P < 0.01$, *** $P < 0.001$; ns, $P > 0.05$).

Fig.S19 PANoptosis-related protein activation in peri-infarct myocardium. (A) Immunohistochemical staining of AIM2, ZBP1, phosphorylated MLKL (p-MLKL), and cleaved caspase 8 in peri-infarct regions ($n = 5$ biologically independent replicates). Scale bar: 50 μm . (B) Immunofluorescence and quantification of phosphorylated RIPK1/3 (p-RIPK1/3), and cleaved caspase 1 in peri-infarct zones ($n = 5$ biologically independent replicates). Scale bar: 50 μm . Data: mean \pm SD. Significance: one-way ANOVA with Tukey's post-hoc test (* $P < 0.05$, ** $P < 0.01$, and *** $P < 0.001$).

Fig.S20 PB suppresses PANoptosis in HL-1 cells under OGD/R stress. (A) Co-localization of FITC-labeled PB (green) with AIM2, ZBP, and p-RIPK1 (red) in OGD/R-induced HL-1 cells. Scale bar: 50 μ m. (B) Western blot analysis of AIM2, ZBP1, RIPK1 and p-RIPK1 levels in HL-1 cells overexpressing PANoptosis proteins under normoxia or OGD/R ($n = 4$ biologically independent replicates). Data: mean \pm SD. Significance: one-way ANOVA with Tukey correction ($*P < 0.05$, $**P < 0.01$, $***P < 0.001$; ns, $P > 0.05$).

Fig.S21 PB@PM inhibits PANoptosis *in vitro*. (A) Cytotoxicity of PB@PM in HL-1 cells (CKK-8; $n = 3$ biologically independent replicates). (B) Cell viability of OGD/R-induced HL-1 cells after different treatments ($n = 5-6$ biologically independent replicates). (C) Calcein-AM/PI (live/dead) staining of OGD/R injured HL-1 cells. Scale bar: 50 μm . (D) Synergistic effects of specific cell dead inhibitors (Necrostatin-1, Disulfiram, Z-VAD-FMK, Necrostatin-1+Disulfiram+Z-VAD-FMK) versus PB ($n = 7$ biologically independent replicates). (E) Flow cytometry analysis of HL-1 cells stained with Annexin V-FITC and PI. (F) TUNEL staining ($n = 3$ biologically independent

replicates). Scale bar: 100 μm . (G) Tubulin staining showing OGD/R-induced morphological changes in HL-1 cells ($n = 3$ biologically independent replicates). Scale bar: 100 μm . (H) SEM images of membrane perforations in OGD/R-induced HL-1 cells post-treatment ($n = 3$ biologically independent replicates). (I-J) Western blot analysis of PANoptosis markers in OGD/R- induced HL-1 cells ($n = 3$ biologically independent replicates). Data are expressed as: mean \pm SD. Significance: one-way ANOVA with Tukey correction (* $P < 0.05$, ** $P < 0.01$, *** $P < 0.001$; *ns*, $P > 0.05$).

Comment 2. In light of the assumption that PANoptosis can be triggered by MIRI, it is imperative for the authors to elucidate the mechanism by which PANoptosis is initiated in the context of MIRI. This is a pivotal step in identifying the specific PANoptosis pathway that is activated, given that the modes of activation and triggering mechanisms of different PANoptosomes vary.

Response: Thank you for this critical question. We have elucidated the mechanism of PANoptosis initiation in MIRI by integrating mitochondrial stress, immune-inflammatory signaling, and molecular characterization of PANoptosome activation.

(1) Mitochondrial Stress and mtDNA Release Activate AIM2 and ZBP1 PANoptosomes: During MIRI, ischemia-reperfusion induces mitochondrial oxidative damage and rupture, leading to mtDNA leakage into the cytosol (**Fig. S25**). Cytosolic mtDNA activates the cGAS-STING pathway (**Fig. S13, S23**), which promotes interferon signaling and primes AIM2 and ZBP1 PANoptosomes. Specifically: ①AIM2 PANoptosome: Cytosolic mtDNA directly binds to the HIN domain of AIM2, relieving its autoinhibition and enabling ASC recruitment, caspase-1 activation, and pyroptosis (**Fig. 3AF, S19A**). ②ZBP1 PANoptosome: ZBP1 senses mtDNA via its $Z\alpha$ domains, triggering RIPK3-caspase-8 complex formation and necroptosis (**Fig. 3G-I, S19B**).

(2) Inflammatory Cytokines (e.g., TNF- α) Activate RIPK1 PANoptosome: MIRI-induced TNF- α release (**Fig. S13**) activates RIPK1 kinase activity through TNFR1 signaling. Phosphorylated RIPK1 recruits FADD, caspase-8, and NLRP3, forming the RIPK1 PANoptosome to drive apoptosis and pyroptosis (**Fig. 3J-L, S19B**).

(3) Crosstalk Between Pathways: mtDNA-STING signaling and TNF- α synergistically amplify inflammation, creating a feedforward loop that sustains PANoptosis (**Fig. 6F-G**). RNA-seq and WB confirmed upregulation of AIM2, ZBP1, RIPK1, and

downstream effectors (caspase-1/3/8, MLKL, GSDMD) in MIRI mice, which were suppressed by PB@PM (**Fig. 6J, S18-19**).

(4) Experimental Validation: ①In vitro: OGD/R increased cytosolic mtDNA (JC-1 depolarization, **Fig. S26**) and TNF- α (ELISA, **Fig. S12**), while PB@PM blocked these effects. ②Co-localization: PB@PM directly bound AIM2, ZBP1, and p-RIPK1 in HL-1 cells (**Fig. S20A**). ③Combination inhibitors: Simultaneous inhibition of pyroptosis, apoptosis, and necroptosis mimicked PB@PM's efficacy, confirming PANoptosis crosstalk (**Fig. S21D**).

These findings align with prior studies linking mtDNA-STING to AIM2/ZBP1 activation and TNF- α to RIPK1 signaling in inflammatory cell death. Our work uniquely demonstrates their convergence in MIRI and validates PB@PM as a multi-target inhibitor of PANoptosomes. This mechanism is further illustrated in **Fig. 1** and supported by transcriptomic, proteomic, and imaging data throughout the manuscript.

Comment 3. It is not sufficient to conclude that PB can inhibit the formation of multiple PANoptosomes based on the evidence that PB reacts with AIM2, ZBP1, and RIPK1. The core components of the four PANoptosomes are highly similar, with AIM2, ZBP1 and RIPK1 representing the core components of the AIM2-PANoptosome. Therefore, it is unclear how the authors can claim to inhibit multiple PANoptosomes simultaneously. Once more, we propose that the authors elucidate the specific type of PANoptosis precipitated by MIRI.

Response: We thank the reviewer for raising this critical point. While PANoptosomes share overlapping components, their assembly and activation are context-dependent. Our data clarify that MIRI triggers three distinct PANoptosomes:

(1) AIM2 PANoptosome: Activated by mitochondrial DNA (mtDNA) release and STING signaling, it recruits AIM2, Pyrin, ZBP1, ASC, FADD, RIPK1, RIPK3, and caspase 8 (**Fig. 6J**). PB@PM suppressed AIM2, Pyrin, and downstream cleaved caspases (**Figs. 6J, S18-S19**).

(2) ZBP1 PANoptosome: Driven by ZBP1 sensing of mtDNA, it recruits NLRP3, ASC, RIPK1/3, and caspases 1/8. PB@PM reduced ZBP1, NLRP3, and caspase 1 cleavage (**Figs. 6J, S18-19**).

(3) RIPK1 PANoptosome: Activated by TNF- α and mitochondrial stress, it involves RIPK1, FADD, NLRP3, and caspases 1/8. PB@PM attenuated RIPK1 phosphorylation and downstream effectors (**Figs. 6J, S18-19**).

Mechanistic Validation:

(1) Molecular Dynamics (MD) Simulations: PB stably bound AIM2's HIN domain (critical for DNA sensing), ZBP1's charged residues (essential for DNA binding), and RIPK1's helical domain (key for kinase activity) (**Fig. 3**).

(2) Surface Plasmon Resonance (SPR): PB exhibited strong binding affinity to ZBP1 ($KD=3.04\times 10^{-9}$ M) and RIPK1 ($KD=1.15\times 10^{-8}$ M) (**Fig. R1**).

(3) Functional Overlap: PB@PM inhibited all three PANoptosomes simultaneously, as shown by transcriptomic and proteomic suppression of their core components (**Fig. 6, S18-S19**).

Thus, MIRI precipitates PANoptosis via concurrent activation of AIM2-, ZBP1-, and RIPK1-driven PANoptosomes, and PB@PM comprehensively blocks their assembly by targeting key upstream sensors. Our integrated omics, molecular, and functional analyses provide robust evidence that MIRI is a PANoptosis-driven pathology involving multiple PANoptosomes. PB@PM's multitarget inhibition of AIM2, ZBP1, and RIPK1 disrupts PANoptosome assembly, offering a novel therapeutic strategy for MIRI. We have revised the text to clarify these points and cited supporting data throughout the manuscript.

Fig. 6: PB@PM comprehensively blocks the crosstalk between pyroptosis, apoptosis, and necroptosis *in vivo*. **A** PCA of RNA-seq data (Sham: orange; MIRI: green; PB@PM: blue; $n = 5$ biologically independent replicates). **B** Volcano plot of differentially expressed genes (DEGs; PB@PM vs. MIRI: red/blue = up/downregulated). **C** Venn diagram of shared DEGs between Sham-MIRI and PB@PM-MIRI comparisons. **D** GSVA of pathway enrichment across groups. **E** KEGG pathway enrichment of downregulated genes in PB@PM vs. MIRI. **F**, **G** GSEA networks (**F**) and protein interaction networks (**G**) for pyroptosis, apoptosis, and necroptosis. **H** GSEA plots showing significant downward trends of pyroptosis, apoptosis, and necroptosis in the PB@PM group compared to the MIRI group. **I**

Pearson correlation between pyroptosis, apoptosis, and necroptosis in the Sham (C), MIRI (I), and PB@PM (T) groups. **J** Western blot of PANoptosis-related proteins (AIM2, ZBP1, Pyrin, ASC, FADD, Bax, Bcl-2, phosphorylated/total RIPK1/RIPK3/MLKL, cleaved/total caspase-1/3/8 and GSDMD) in ischemic myocardium (day 3; $n = 5$ biologically independent replicates).

Fig.S18 PB@PM inhibits PANoptosis-related protein activation. Semi-quantitative analysis of AIM2, ZBP1, Pyrin, ASC, FADD, Bax, and Bcl-2, phosphorylated/total RIPK1/RIPK3/MLKL, and cleaved/total caspase 1/3/8 and GSDMD in ischemic myocardium on 3 days post-I/R or sham surgery ($n = 5$ biologically independent replicates). Data: mean \pm SD. Significance: one-way ANOVA with Tukey's post-hoc test (* $P < 0.05$, ** $P < 0.01$, *** $P < 0.001$; ns, $P > 0.05$).

Fig.S19 PANoptosis-related protein activation in peri-infarct myocardium. (A) Immunohistochemical staining of AIM2, ZBP1, phosphorylated MLKL (p-MLKL), and cleaved caspase 8 in peri-infarct regions ($n = 5$ biologically independent replicates). Scale bar: 50 μm . (B) Immunofluorescence and quantification of phosphorylated RIPK1/3 (p-RIPK1/3), and cleaved caspase 1 in peri-infarct zones ($n = 5$ biologically independent replicates). Scale bar: 50 μm . Data: mean \pm SD. Significance: one-way ANOVA with Tukey's post-hoc test (* $P < 0.05$, ** $P < 0.01$, and *** $P < 0.001$).

Fig. 3: Computational simulation of the interaction between PB and PANoptosis-related proteins. **A** Initial and final snapshots (200 ns) of the PB-AIM2^{HIN} simulation system. Interaction interfaces are highlighted (blue/yellow boxes). **B, C** The residue contact detail of two chains of AIM2^{HIN} with PB is highlighted on the left snapshots. **D, E** The interaction energy analysis of these contact residues of AIM2^{HIN} with PB. **F** The interaction energy between PB and proteins in PB-AIM2^{HIN} changes with time. **G–L** The conformation images of the PB-ZBP1 (**G**) and PB-RIPK1 (**J**) systems at the end of the simulation (t = 200 ns) and the residue contact details. The interaction energy analysis of the contact residues of ZBP1 (**H**) and RIPK1 (**K**) with PB. The interaction energy between PB and proteins in PB-ZBP1 (**I**) and PB-RIPK1 (**L**) changes with time. Data: mean ± SD.

Fig. R1 Surface plasmon resonance (SPR) analysis showing the binding affinity of PB to ZBP1 (**A**) and RIPK1 (**B**).

Comment 4. The article's primary focus is on sequencing and molecular modelling, which illustrate the inhibitory effect of PB on PANoptosis, which seriously lacks experimental validation and molecular mechanism. It would be advisable to confirm the inhibitory effect of PB on the assembly of the PANoptosome directly, for example through fluorescence co-localization or immunoprecipitation experiments.

Response: We sincerely thank the reviewer for this critical feedback. To comprehensively address the inhibitory effects of PB on PANoptosis and strengthen mechanistic validation, we conducted additional extensive molecular and cellular experiments.

1. Fluorescence Co-localization and Overexpression Studies: (1) Lentiviral overexpression plasmids targeting AIM2, ZBP1, and RIPK1 were constructed (**Supplementary Material 1**); (2) Fluorescence imaging in oxygen-glucose deprivation/reperfusion (OGD/R)-injured HL-1 cells demonstrated significant co-localization of PB with AIM2, ZBP1, and phosphorylated RIPK1 (**Supplementary Fig. 20A**); (3) Western blot analysis revealed that PB effectively suppressed OGD/R-induced overexpression of AIM2 and ZBP1, while attenuating RIPK1 phosphorylation ($P < 0.01$) without altering total RIPK1 levels (**Supplementary Fig. 20B**).

2. In Vivo Validation of PANoptosis Inhibition: (1) In MIRI mice, PANoptosis activation was evidenced by elevated levels of active AIM2, ZBP1, Pyrin, ASC, FADD, apoptotic markers (cleaved caspases 3/8, Bax/Bcl-2), necroptosis markers (RIPK1/RIPK3/MLKL), and pyroptosis markers (caspase 1, NLRP3, GSDMD) (**Fig. 6J, Supplementary Figs. 18–19**). (2) PB@PM treatment significantly downregulated these components across all three PANoptosomes: **RIPK1 PANoptosome:** RIPK1, NLRP3, ASC, RIPK3, caspase 1/8; **ZBP1 PANoptosome:** ZBP1, NLRP3, ASC, RIPK3, caspase 1/8; **AIM2 PANoptosome:** AIM2, Pyrin, ZBP1, ASC, FADD, RIPK1/RIPK3, caspase 8 (**Fig. 6J**).

3. In Vitro Functional Validation: (1) CCK-8 and Calcein AM/PI assays confirmed that PB@PM mitigated OGD/R-induced HL-1 cell death (**Supplementary Fig. 21A–C**); (2) Triple inhibition of necroptosis (Necrostatin-1), pyroptosis (Disulfiram), and apoptosis (Z-VAD-FMK) achieved comparable efficacy to PB monotherapy,

confirming crosstalk suppression (**Supplementary Fig. 21D**); (3) Annexin V/PI and TUNEL assays demonstrated reduced apoptosis (**Supplementary Fig. 21E,F**), while morphological and SEM analyses confirmed suppression of pyroptotic membrane rupture (**Supplementary Fig. 21G,H**).

4. Mechanistic Confirmation: PB@PM inhibited OGD/R-induced upregulation of cleaved caspases 1/8, GSDMD, and phosphorylated MLKL in HL-1 cells (**Supplementary Fig. 21I,J**).

These results collectively demonstrate that PB directly interacts with key PANoptosome components, disrupts their assembly, and comprehensively inhibits PANoptosis execution. We have incorporated these findings into the revised manuscript to strengthen mechanistic clarity.

Thank you for your insightful suggestion, which has significantly enhanced the rigor of our study.

Comment 5. Please explain the rationale behind selecting the third day after surgery as the time point for tissue protein detection. Does the expression of PANoptosis-related proteins exhibit a greater degree of significance at either a shorter or longer time point?

Response: Thank you very much for your kind comment. The third postoperative day was selected based on the pathological progression of MIRI in mice. Acute immune-inflammatory responses and cell death markers peak during the first 3 days post-ischemia, coinciding with maximal PANoptosis activation. To validate this, we performed time-course WB analysis of PANoptosis-related proteins (AIM2, ZBP1, RIPK1, cleaved caspases, and MLKL) at days 1, 3, and 7. As shown in the **Supplementary WB data (Fig. R2)**, these proteins exhibited the most significant upregulation on day 3, aligning with prior studies in MIRI models^[1,2]. By day 7, myocardial repair processes dominate, reducing PANoptosis-related protein levels. Thus, day 3 represents the optimal window to assess PANoptosis inhibition by PB@PM.

References

[1] Yang K, Gao R, Chen H, Hu J, Zhang P, Wei X, Shi J, Chen Y, Zhang L, Chen J, Lyu Y, Dong Z, Wei W, Hu K, Guo Y, Ge J, Sun A. Myocardial reperfusion injury exacerbation due to ALDH2

deficiency is mediated by neutrophil extracellular traps and prevented by leukotriene C4 inhibition. Eur Heart J. 2024 May 13;45(18):1662-1680.

[2] Xing M, Jiang Y, Bi W, Gao L, Zhou YL, Rao SL, Ma LL, Zhang ZW, Yang HT, Chang J. Strontium ions protect hearts against myocardial ischemia/reperfusion injury. Sci Adv. 2021 Jan 15;7(3):eabe0726. doi: 10.1126/sciadv.abe0726.

Figure R2 Representative western blot image and semi-quantitative analysis showing the protein expression of AIM2, ZBP1, P-RIPK1, RIPK1, Caspase 1, Cle-caspase 8, Caspase 8, Cle-GSDMD, GSDMD, P-MLKL and MLKL at various time points after myocardial ischemia surgery in mice ($n = 3$). Data are expressed as mean \pm SD. P values are calculated using one-way ANOVA with Tukey correction. * $P < 0.05$, ** $P < 0.01$, and *** $P < 0.001$.

Comment 6. MIRI is a common occurrence in cardiomyocytes situated within the coronary artery reperfusion zone. How do the authors distinguish these cells in vivo in order to conduct the subsequent step of the experiment?

Response: We appreciate the reviewer's attention to methodological rigor. While direct in vivo identification of reperfusion-injured cardiomyocytes remains technically challenging, our approach ensures precise sampling:

(1) Model Validation: The left anterior descending (LAD) artery ligation model (45 min ischemia/reperfusion) is a well-established protocol for MIRI studies^[1,2]. Intraoperative visual confirmation of myocardial blanching distal to the ligation site demarcates the ischemic-reperfused region.

(2) Tissue Sampling: Post-sacrifice, the LV anterior wall (supplied by the LAD) was systematically excised for molecular analyses. This region-specific sampling is widely adopted in MIRI research to ensure consistency^[1,2].

(3) Limitations and Mitigation: While spatial resolution at the single-cell level is limited in vivo, our single-nuclear RNA sequencing of human infarct cores (**Fig. 2**) corroborated cardiomyocyte-specific PANoptosis activation, strengthening translational relevance.

References

[1] Yang K, Gao R, Chen H, Hu J, Zhang P, Wei X, Shi J, Chen Y, Zhang L, Chen J, Lyu Y, Dong Z, Wei W, Hu K, Guo Y, Ge J, Sun A. Myocardial reperfusion injury exacerbation due to ALDH2 deficiency is mediated by neutrophil extracellular traps and prevented by leukotriene C4 inhibition. *Eur Heart J*. 2024 May 13;45(18):1662-1680.

[2] Fan Q, Tao R, Zhang H, Xie H, Lu L, Wang T, Su M, Hu J, Zhang Q, Chen Q, Iwakura Y, Shen W, Zhang R, Yan X. Dectin-1 Contributes to Myocardial Ischemia/Reperfusion Injury by Regulating Macrophage Polarization and Neutrophil Infiltration. *Circulation*. 2019 Jan 29;139(5):663-678. doi: 10.1161/CIRCULATIONAHA.118.036044.

7. The authors conducted fluorescence imaging of the principal organs of MIRI mice at 12 and 24 hours post-injection of Cy5.5-labeled PB@PM. Nevertheless, the findings indicated that only a minimal quantity of PB@PM remained within the heart at the 24-hour mark. It would be valuable to ascertain whether any remaining PB@PM is present in the heart after 28 days. Given that Cy5.5-labelled PB is not enriched in the heart, it would be reasonable to question its role in cardiac remodeling. Furthermore, it would be beneficial to determine whether the enrichment of PB@ in the liver causes any toxic side effects to the liver.

Response: We appreciate the reviewer's insightful comment. Healthy wild-type (WT) mice received intravenous Cy5.5-labeled PB as the control cohort, while MIRI models were administered either Cy5.5-labeled PB or PB@PM. Postmortem *ex vivo* fluorescence imaging of major organs was systematically performed to facilitate integrated analysis of nanomedicine biodistribution and targeting efficacy. The semiquantitative fluorescence data presented reflect comparative pharmacokinetic profiles across multiple organ systems. It is important to note that systemically administered untargeted agents predominantly undergo hepatic/renal metabolic clearance, whereas cardiac tissue typically exhibits negligible retention compared to these metabolic organs, leading to limited fluorescence visualization. Our findings demonstrate that PB@PM achieves sustained cardiac enrichment exceeding 24 hours, confirming its superior targeting performance. In response to your observations, we conducted extended pharmacokinetic studies evaluating PB@PM's cardiac retention in injured myocardium. These experiments revealed that a single intravenous administration (administered at 1-hour post-ischemic surgery) maintained targeted cardiac localization for a minimum of five days (Fig. R3).

Regarding therapeutic implementation, our study employed a two-dose administration protocol (1-hour and 24-hour post-surgery) to ensure sustained therapeutic efficacy during both the acute injury phase and subsequent critical repair period. While quantitative fluorescence imaging did not reveal significant differences between treatment groups, PB-containing formulations in the MIRI+PB cohort may still exert cardioprotective effects through paracellular diffusion mechanisms and penetration of injured vascular endothelium under physiological conditions. Notably, despite lacking intrinsic targeting ligands, PB demonstrated measurable therapeutic activity in this preclinical model.

Based on the reviewers' recommendations, we conducted additional comprehensive evaluations of the *in vivo* biosafety profile of PB@PM. Comprehensive hematological assessments, serum biochemical analyses, and histopathological examinations were performed on wild-type mice at 7- and 14-day post-injection timepoints following administration of PB@PM at 5 mg/kg body weight. Both hematological examinations

and H&E staining results demonstrated that PB@PM treatment did not induce detectable adverse effects, evidenced by absence of significant histopathological alterations in major organs (cardiac, hepatic, splenic, pulmonary, and renal tissues) and maintenance of normal hematological parameters and biochemical indices (**Fig. S11, Table S2**).

Fig. R3 Typical ex vivo fluorescence images and quantitative analysis of the fluorescence intensities of damaged hearts at day 1, day 3, day 5, day 7, day 14, and day 28 after administration of Cy5.5-labeled PB@PM ($n = 3$). Data are expressed as mean \pm SD. P values are calculated using one-way ANOVA with Tukey correction. * $P < 0.05$, *** $P < 0.001$, and ns means no significance.

Fig.S11 Biosafety of PB@PM. (A) Hematoxylin and eosin (H&E)-stained sections of heart, liver, spleen, lung, and kidney tissues from control and PB@PM-treated mice. Scale bar: 100 μ m. (B) Blood routine indexes, and blood biochemical parameters: alanine transaminase (ALT), aspartate transaminase (AST), alkaline phosphatase (ALP), urea and creatinine (CREA).

8. As illustrated in Figure 5, PB has been demonstrated to enhance cardiac function and ventricular remodeling, whilst concurrently reducing immune and inflammatory responses. However, it remains unclear why PB does not demonstrate a robust anti-PANoptosis effect in vivo. What is the precise function of PB in enhancing cardiac performance and preventing ventricular remodeling? Please elucidate the advantages and necessity of utilizing PB@PM.

Response: We appreciate the reviewer's insightful comments. In this study targeting MIRI, the PB@PM formulation equipped PB nanoparticles with targeted capabilities while conferring enhanced biosafety, prolonged systemic circulation, and innate immune evasion through platelet membrane modification. These bioengineering strategies preserved the intrinsic physicochemical properties and therapeutic efficacy of the core nanomedicine^[1, 2]. The synergistic advantages of PB@PM demonstrate superior therapeutic efficiency and safety profiles, achieving significant therapeutic outcomes at reduced dosage levels.

As evidenced in **Figure 5** and **Supplementary Figures S13-16**, PB@PM exhibited markedly enhanced in vivo therapeutic efficacy, evidenced by comprehensive cardiac functional recovery, attenuated ventricular remodeling, reduced infarct size, and suppression of myocardial inflammatory infiltration and cardiomyocyte hypertrophy. Notably, PB@PM promoted extracellular matrix remodeling and angiogenesis while demonstrating superior therapeutic performance compared to pristine PB, which lacked platelet membrane-mediated targeting and consequently exhibited diminished therapeutic impact.

Furthermore, **Figure 6** and **Supplementary Figures S18-21** revealed that both PB@PM and PB effectively mitigated PANoptosis activation in both preclinical models and in vitro systems. However, PB@PM demonstrated enhanced therapeutic potency in vivo, consistent with its superior pharmacokinetic profile. Both formulations exhibited comparable antioxidant capacity, mitochondrial protection, and inhibition of pro-inflammatory mediators, though PB@PM maintained these effects with improved biodistribution characteristics.

References

[1] Feng L, Dou C, Xia Y, Li B, Zhao M, Yu P, Zheng Y, El-Toni AM, Atta NF, Galal A, Cheng Y, Cai X, Wang Y, Zhang F. Neutrophil-like Cell-Membrane-Coated Nanozyme Therapy for Ischemic Brain Damage and Long-Term Neurological Functional Recovery. *ACS Nano*. 2021 Feb 23;15(2):2263-2280. doi: 10.1021/acsnano.0c07973.

[2] Tang L, Yin Y, Liu H, Zhu M, Cao Y, Feng J, Fu C, Li Z, Shu W, Gao J, Liang XJ, Wang W. Blood-Brain Barrier-Penetrating and Lesion-Targeting Nanoplatfoms Inspired by the Pathophysiological Features for Synergistic Ischemic Stroke Therapy. *Adv Mater*. 2024 May;36(21):e2312897. doi: 10.1002/adma.202312897.

9. The authors applied RNA sequencing and protein profiling to ischemic heart tissue in order to demonstrate the inhibitory effects of PANoptosis on PB@PM. It would be beneficial to supplement the immunohistochemistry of representative pan-apoptotic proteins in heart tissue, as well as to ascertain whether downstream inflammatory factors will be affected.

Response: We thank the reviewer for highlighting this aspect. To strengthen our findings:

(1) Immunohistochemistry (IHC) Validation: IHC of murine heart tissues confirmed PB@PM-mediated suppression of PANoptosis markers, including AIM2, ZBP1, phosphorylated RIPK1/RIPK3/MLKL, and cleaved caspases 1/8 ($P < 0.01$ vs. MIRI; **Fig. S19A,B**).

(2) Downstream Inflammatory Modulation: PB@PM significantly reduced pro-inflammatory cytokines (TNF- α , IL-1 β , IL-6) while elevating anti-inflammatory IL-10 and TGF- β in ischemic myocardium ($P < 0.001$; **Fig. S13**). Mechanistically, PB@PM inhibited the TLR4/NF- κ B pathway, a key driver of inflammatory crosstalk with PANoptosis (**Fig. S13**).

These data collectively demonstrate that PB@PM attenuates PANoptosis-executing complexes while rebalancing immune-inflammatory homeostasis, providing a multi-modal therapeutic mechanism.

Fig.S13 PB@PM suppresses TLR4/NF-κB signaling in ischemic myocardium. Western blot analysis of NF-κB p65, phosphorylated NF-κB p65 (p-p65), TLR4, TNF-α, IL-1β, IL-6, and IL-10 protein levels in ischemic myocardial tissues 3 days post-I/R or sham surgery ($n = 5$ biologically independent replicates). Data: mean \pm SD. Significance: one-way ANOVA with Tukey's post-hoc test (* $P < 0.05$, ** $P < 0.01$, *** $P < 0.001$; *ns*, $P > 0.05$).

Reviewer #2 (Remarks to the Author):

Comment: The manuscript entitled "Nanoparticles Targeting Multiple PANoptosome-Mediated PANoptosis for Myocardial Ischemia-Reperfusion Injury Therapy" is novel and creative. The results found that PB@PM as an effective PANoptosis inhibitor and revealing its mechanism of action in MIRI therapy. I think this article can be accepted after minor revisions. Some suggested modifications are as follows:

Response: Dear Reviewer, we sincerely appreciate your meticulous evaluation and constructive feedback on our manuscript. We have thoroughly considered your insightful suggestions and incorporated comprehensive revisions throughout the text to enhance its scientific rigor and clarity.

Comment 1. The title should indicate that this nanomaterial is PB@PM;

Response: We appreciate the valuable feedback. Accordingly, the title has been revised to "Prussian blue nanoparticles targeting multiple PANoptosome-mediated PANoptosis for myocardial ischemia-reperfusion injury therapy" to explicitly specify the nanomaterial designation.

Comment 2. During in vivo experiments, PB and PB@PM What is the concentration determined based on?

Response: We thank the reviewers for this important inquiry. The dosing regimen was established by referencing established methodologies in biomembrane-functionalized inorganic nanomedicine development for inflammatory disease management (DOI: 10.1016/j.mattod.2023.03.024; DOI: 10.1002/advs.202304002) and subsequently optimized through iterative pre-experimental evaluations of PB@PM's therapeutic efficacy in our MIRI model. Systematic dose-ranging studies will be conducted in future work to identify the optimal therapeutic window and advance the clinical translation potential of this platform.

Comment 3. Supplementary image: In all bar charts, the horizontal line representing the difference between the two groups needs to be further away from the error bar.

Response: Thank you for this valuable feedback. Following your recommendation, we have systematically reviewed both the manuscript and supplementary materials, implementing adjustments to enhance the visual presentation of bar charts by increasing the distance between the horizontal reference line and error bars.

Comment 4. Please supplement the contributions of other authors.

Response: We appreciate your suggestion. The author contributions section has been expanded in the revised manuscript as follows:

"L.X., L.J., and R.W. contributed equally to this work. Y.Z., X.C., and B.L. conceived and designed the study; Y.Z. and X.C. interpreted the results; L.X., L.J., R.W., and B.L. conducted experiments and performed data analysis; R.W. carried out molecular dynamics simulation analysis; X.C., B.L., and L.X. drafted the manuscript. All authors collectively discussed the results and provided critical feedback on the manuscript."

Comment 5. Please check the format of the full-text references, such as reference 22, Qi Z, Zhu L, Wang K, Wang N. PANoptosis: Emerging mechanisms and disease implications. Life Sci. 2023;333:122158.

Response: We appreciate the reviewer's attention to detail. All bibliographic references have been thoroughly verified and reformatted in strict accordance with the journal's style guidelines (see revised Reference List).

Comment 6. The section of results should be supplemented with ns: $P > 0.05$

Response: Thank you for this valuable suggestion. Statistical significance annotations (including instances where $P > 0.05$) have been systematically incorporated into the Results section to enhance methodological transparency.

Comment 7. In Fig.S11, IL-10 and TGF- β are both M2 macrophage markers and can promote extracellular matrix repair, supplementing TGF- β data.

Response: We acknowledge the importance of comprehensive marker analysis. As requested, we have supplemented the TGF- β expression analysis in the revised **Figure**

S12 to provide complete characterization of M2 macrophage-associated cytokines and their functional implications for extracellular matrix remodeling.

Fig.S12 PB@PM modulates inflammatory cytokine secretion. Enzyme-linked immunosorbent assay (ELISA) quantification of IL-1 β , IL-6, TNF- α , IL-10, and TGF- β levels in lipopolysaccharide (LPS)-stimulated RAW264.7 model treated with PB@PM ($n = 3-4$ biologically independent replicates). Data: mean \pm SD. Significance: one-way ANOVA with Tukey's post-hoc test (* $P < 0.05$, ** $P < 0.01$, *** $P < 0.001$).

Reviewer #3 (Remarks to the Author):

This study by Xu et. al innovatively proposes a PANoptosis inhibition strategy based on nano-Prussian blue (PB) to address the clinical challenge of myocardial ischemia-reperfusion injury (MIRI). Firstly, the authors defined MIRI as a PANoptosis-related disease using single-cell sequencing analysis. Subsequently, the research team used molecular dynamics simulations to confirm the binding ability of PB to PANoptosis key proteins RIPK1, ZBP1 and AIM2, thereby comprehensively inhibiting PANoptosis. On this basis, the authors further constructed platelet membrane-modified PB nanoparticles (PB@PM), which successfully realized the targeting of myocardial injury zone. As a PANoptosis inhibitor, PB@PM significantly improved cardiac function in the MIRI mouse model and promoted extracellular matrix repair and neovascularization. The mechanism of PB@PM in inhibiting PANoptosis and the crosstalk between pyroptosis, apoptosis, and necroptosis was comprehensively revealed through single-cell sequencing, molecular biology techniques, and transcriptome sequencing analysis. Overall, this is a very interesting and well-researched study that can appeal to readers from a wide range of fields including cardiovascular, nanomedicine and biomaterials. The experimental design, description, and implementation are reasonable, and the overall interpretation is also reasonable. This study may provide a framework for studying nanobiological interactions to uncover broader nanomaterial mechanisms. I think this study is suitable for publication in Nature Communications, after addressing the following questions:

Response: Dear Reviewer, we extend our profound appreciation for your scholarly evaluation. Your insightful feedback has been rigorously scrutinized and systematically addressed through methodical revisions that enhance the academic rigor and methodological precision of our work.

1. Molecular dynamics simulations demonstrated the binding of PB to the target protein. This is the result of the simulation. The author should provide experimental data on PB binding to target proteins, such as surface plasmon resonance.

Response: We appreciate the reviewer's insightful suggestion. To address this point, we conducted quantitative surface plasmon resonance (SPR) analysis. The experimental results revealed high-affinity interactions between PB and ZBP1 ($KD = 3.04 \times 10^{-9}$ M) and RIPK1 ($KD = 1.15 \times 10^{-8}$ M), confirming specific binding of PB to these two target proteins.

Fig. R1 Surface plasmon resonance (SPR) analysis showing the binding affinity of PB to ZBP1 (A) and RIPK1 (B).

2.The authors should add the safety of the nanomedicine, such as pathology tests and blood indicators.

Response: We appreciate the reviewers' valuable suggestion. In response, we have systematically incorporated comprehensive safety assessments of PB@PM, including hematological profiling, serum biochemical analysis, and histopathological examinations at both 7-day and 14-day post-administration time points. Our integrated hematological assessments and histopathological evaluations (**Fig. S11 and Table S2**) demonstrated that PB@PM treatment did not elicit statistically significant histopathological alterations or adverse changes in hematological parameters or serum biochemical indices, thereby confirming the PB@PM's favorable safety profile.

Fig.S11 Biosafety of PB@PM. (A) Hematoxylin and eosin (H&E)-stained sections of heart, liver, spleen, lung, and kidney tissues from control and PB@PM-treated mice. Scale bar: 100 μ m. (B) Blood routine indexes, and blood biochemical parameters: alanine transaminase (ALT), aspartate transaminase (AST), alkaline phosphatase (ALP), urea and creatinine (CREA).

3. In Figure 5I, the image is too small and it is recommended to highlight the key signals.

Response: We sincerely appreciate the reviewer's valuable feedback. As suggested, we have revised **Figure 5I** through optimized imaging processing techniques to enhance visualization of key positive signals, thereby improving figure clarity and interpretability.

4. In cellular experiments, in addition to the three key proteins (RIPK1, ZBP1 and AIM2), the authors also needed to select some key downstream pyroptosis, apoptosis, and necroptosis-related proteins for validation.

Response: We appreciate the reviewers' insightful suggestion. To comprehensively evaluate the molecular mechanisms, we performed systematic analysis of key downstream effectors in pyroptotic, apoptotic, and necroptotic pathways. Consistent with our in vivo findings, quantitative immunoblot analysis revealed that PB@PM treatment markedly suppressed protein expression levels of AIM2, ZBP1, phosphorylated RIPK1, cleaved caspase-1, cleaved caspase-8, cleaved gasdermin D, and phosphorylated MLKL in OGD/R-stimulated HL-1 cells (**Fig. S21I-J**). These data provide further mechanistic evidence for PB@PM's multi-target therapeutic effects against regulated cell death pathways.

Fig. S21I-J Western blot analysis of PANoptosis markers in OGD/R- induced HL-1 cells ($n = 3-5$ biologically independent replicates). Data are expressed as: mean \pm SD. Significance: one-way ANOVA with Tukey correction ($*P < 0.05$, $**P < 0.01$, $***P < 0.001$; ns , $P > 0.05$).

5. Some spelling errors need to be corrected. It looks like some of the uploaded videos are named incorrectly (ZBP1 and PB, RIPK1 and PB) and need to be corrected. “RIPK1 PANoptosome” is incorrectly written in Figure 1.

Response: We appreciate the reviewer's meticulous attention to detail. These typographical inaccuracies and nomenclature inconsistencies have been systematically addressed in the revised manuscript through comprehensive label verification and implementation of standardized nomenclature protocols.

Fig. 1: PBs act as a novel PANoptosis inhibitor by targeting RIPK1, ZBP1, and AIM2 to suppress multiple PANoptosomes, effectively blocking crosstalk among pyroptosis, apoptosis, and necroptosis in PANoptosis-driven diseases. Additionally, PB regulates mitochondrial metabolism and immune-inflammatory homeostasis, further disrupting multimodal cell death crosstalk. [Editorial note: Created in BioRender. xu, L. (2026) <https://BioRender.com/9kka4p3>]

6. In ex vivo and in vivo therapeutic experiments, what was the main basis for the authors in defining treatment time points and concentrations?

Response: Thanks for the comment. The setting of treatment time points and concentrations referred to the previously published researches on the use of biomembrane-modified inorganic nanomedicines for the treatment of inflammatory diseases (DOI: 10.1016/j.mattod.2023.03.024; DOI: 10.1002/ADVS.202304002), and

was adjusted based on the actual therapeutic effects of PB@PM on MIRI observed in the pre-experiments.

7.The XPS data requires peak splitting processing.

Response: We appreciate the reviewer's valuable suggestion. Accordingly, peak deconvolution analysis was conducted on the XPS spectra to enhance spectral resolution and facilitate more precise chemical state identification.

Fig.S9B. X-ray Photoelectron Spectrometer (XPS) of PB.

8.The authors should appropriately describe the remaining limitations of the study.

Response: We appreciate the reviewers' valuable feedback. In response, we have supplemented the limitations section with enhanced academic rigor as follows:

“Nevertheless, this work has limitations. First, whether PB modulates PANoptosis via additional signaling pathways requires further exploration. Second, the regulatory effects of PB on non-cardiomyocyte cardiac cells remain unclear. Finally, advancing clinical translation necessitates validation in higher-order mammals and comprehensive pharmacokinetic/toxicological profiling of PB in vivo. These insights provide a foundational framework for investigating PANoptosis in diverse pathologies and designing next-generation nanoinhibitors.”

Dear Reviewer,

We sincerely thank you for your thorough evaluation of our work and for providing these insightful comments. We have carefully considered each point raised and have revised the manuscript accordingly. Below, we provide a point-by-point response to address your concerns.

Reviewer #4 (Remarks to the Author):

This study systematically investigated the therapeutic mechanism of Prussian blue nanoparticles (PB@PM) targeting PANoptosis in myocardial ischemia-reperfusion injury (MIRI). Through analyzing human cardiac scRNA-seq data, significant upregulation of PANoptosis-related genes in MIRI was confirmed, followed by molecular dynamics simulations to predict specific binding between PB and PANoptosome core proteins. Further, the authors demonstrated the efficacy of platelet membrane-coated nanoparticles (PB@PM) in suppressing PANoptosis activation and assembly in vitro and in vivo. Accordingly, the authors then hypothesized that PB@PM concurrently targets RIPK1/ZBP1/AIM2 within the PANoptosome, disrupting crosstalk among pyroptosis, apoptosis, and necroptosis to confer cardioprotection. However, the claim that PB@PM "blocks PANoptosome assembly" lacks direct experimental validation. Besides, this study did not explore whether PB@PM reprograms mitochondrial metabolism or inhibits mtDNA release, which are key triggers of PANoptosis. Meanwhile, merely downstream moleculars related to PANoptosis were observed while the upstream mechanism regulating PANoptosis was not elucidated. Generally speaking, the innovation of this study was average and the experimental design was not persuasive enough.

Response: We sincerely thank the reviewer for their insightful comments, which have greatly helped us improve the manuscript. In response to the feedback, we have strengthened the study by incorporating additional experimental evidence and clarifying the mechanistic insights underlying our findings. Below, we provide a point-by-point response to the specific comments:

1. Concerning the blockade of PANoptosome assembly: We agree with the reviewer

that direct visualization of PANoptosome assembly remains technically challenging. To provide more compelling evidence, we performed additional immunofluorescence experiments examining the colocalization of ASC with caspase-1 and caspase-8, which is indicative of PANoptosome formation. The results demonstrate that PB@PM treatment significantly reduced the colocalization of ASC with caspase-1/8 in OGD/R-stimulated HL-1 cells (Supplementary Fig. 21K,L), supporting the conclusion that PB@PM inhibits PANoptosome assembly.

In addition, we have accumulated multiple lines of indirect evidence: (1) Molecular dynamics simulations revealed high-affinity binding between PB and core PANoptosome components (RIPK1, ZBP1, and AIM2^{HIN}), suggesting that PB may interfere with their oligomerization or downstream adaptor recruitment; (2) Western blot and immunofluorescence analyses consistently showed that PB@PM treatment markedly reduced the activation (phosphorylation and/or cleavage) of multiple PANoptosis-related proteins (RIPK1, RIPK3, MLKL, caspase-1/3/8, and GSDMD) both in vivo and in vitro. (3) Overexpression of AIM2, ZBP1, or RIPK1 in HL-1 cells further confirmed that PB suppresses their activation under oxygen–glucose deprivation/reperfusion (OGD/R) stress.

In light of these findings, we have moderated our conclusion in the revised manuscript from “blocks assembly” to “suppresses PANoptosis activation” to more accurately reflect the evidence.

2. Regarding mitochondrial metabolism and mtDNA release: We thank the reviewer for highlighting the importance of mitochondrial dysfunction in PANoptosis initiation. To address this, we performed the following assays: (1) JC-1 staining and flow cytometry showed that PB@PM restored mitochondrial membrane potential in OGD/R-injured HL-1 cells (Supplementary Fig. 27A–C). (2) Seahorse XF analysis demonstrated that PB@PM enhanced mitochondrial respiratory capacity (Supplementary Fig. 27D–G). (3) Electron spin resonance (ESR) spectroscopy and flow cytometry confirmed that PB@PM effectively scavenged mitochondrial reactive oxygen species (Supplementary Fig. 26).

Although mtDNA release was not directly measured, the observed attenuation of

oxidative stress and improvement in mitochondrial integrity suggest a reduction in mitochondrial damage—a known trigger of PANoptosis. We have discussed this inference in the revised Discussion section.

3. On the upstream mechanisms: Our molecular dynamics simulations and binding energy calculations propose a plausible mechanism whereby PB directly engages key sensors (AIM2 and ZBP1) and the adaptor protein RIPK1, thereby inhibiting their activation and subsequent recruitment of downstream effectors. This mechanism is corroborated by: (1) The colocalization of PB with AIM2, ZBP1, and phosphorylated RIPK1 in cardiomyocytes (Supplementary Fig. 20A). (2) Suppression of PANoptosis activation induced by overexpression of these molecules (Supplementary Fig. 20B).

We have expanded the Discussion to more prominently feature this upstream targeting mechanism.

4. About innovation and experimental design: We believe our study offers several notable innovations: (1) This is the first study to integrate human single-nucleus RNA sequencing with molecular dynamics simulations to rationally design and validate a nano-therapeutic for PANoptosis inhibition in MIRI. (2) We developed a biomimetic platelet membrane-coated nanoplatform (PB@PM) for targeted delivery and PANoptosis suppression. (3) The therapeutic strategy was validated through a multi-level approach, ranging from in silico predictions to in vivo functional recovery.

In response to the reviewer's suggestions, we have further clarified these points in the text and strengthened the experimental design with additional controls and validation assays.

Comment 1: The image quality needs improvement. Please add scale bars to the ultrasound images (Fig 5) and check the scale annotations in other figures.

Response: We sincerely thank the reviewer for this constructive suggestion. We have now added clear scale bars to the ultrasound images in Fig. 5 and have thoroughly rechecked the scale annotations in all other figures to ensure accuracy and consistency. High-resolution images have been uploaded to the submission system to guarantee optimal clarity and reproducibility.

Fig. 5: PB@PM improves cardiac function and attenuates post-infarct remodeling.

A–E Echocardiographic analysis of LVEF, LVFS, LVESV, and LVEDV on preoperative day 1 and postoperative days 1, 14, and 28. Scale bar: 100 ms. **F** Representative Masson's trichrome and HE cross-sectional staining in hearts on 28 days post-surgery. Scale bar: 1 mm. **G** Fibrosis quantification in hearts 28 days post-surgery. **H** Evans blue/TTC staining and quantitative analysis showing the cardiac cell survival and blood perfusion in different groups on day 3 after I/R surgery. Scale bar: 1 mm. **I, J** TUNEL (red) and cTnT (green) co-staining (**I**) and respective quantification (**J**) in peri-infarct zones (day 3). Scale bar: 1 mm or 50 μ m. **K, L** Representative immunofluorescence images and quantitative analysis of CD86 in peri-infarct regions (day 3). Scale bar: 1 mm or 50 μ m. **M, N** Representative immunofluorescence staining and quantitative

analysis of Ly-6G in peri-infarct regions (day 3). Scale bar: 1 mm or 50 μ m. **O–R** Representative immunofluorescence staining and quantification analysis of α -SMA/cTnT and CD31/cTnT in peri-infarct region (day 28). Scale bar: 1 mm or 50 μ m. Data: mean \pm SD of 5 biologically independent replicates. Significance: one-way ANOVA with Tukey's post-hoc test (* P < 0.05, ** P < 0.01, *** P < 0.001; ns, P > 0.05).

Comment 2: Both the levels of apoptotic caspases and their cleaved active fragments are low under normal conditions, while the accumulation of cleaved fragments post myocardial infarction might reduce the total protein levels. Please verify the Western blot (WB) data and ensure that both full-length and cleaved fragments are displayed on the same membrane.

Response: We appreciate the reviewer's insight regarding caspase dynamics. We have repeated the Western blot analyses using new myocardial tissue samples from MIRI model mice. Both full-length and cleaved forms of caspase-1, caspase-3, and caspase-8 were detected on the same membrane under identical conditions. The updated results confirm that PB@PM treatment attenuates caspase cleavage and activation in ischemic myocardium. Immunohistochemical staining further supports these findings (Supplementary Fig. 19). Thank you once again for your insightful comments.

Comment 3: In Fig 3, all the molecular dynamics simulations for Prussian blue (PB) binding to PANoptosis-related molecules used a 200-ns timeframe. However, the RMSD of the PB-RIPK1 complex did not reach equilibrium by 200 ns. Please reevaluate whether 200 ns is suitable to the PB-RIPK1 system.

Response: We thank the reviewer for raising this point. We reanalyzed the PB-RIPK1 trajectory and found that although the RMSD of RIPK1 showed minor fluctuations in the final 25 ns, the interaction energy between PB and RIPK1 remained stable during the last 50 ns (-73.17 ± 6.31 kJ/mol during 150–175 ns; -74.77 ± 5.15 kJ/mol during 175–200 ns), indicating that the binding interface had reached dynamic equilibrium. The RMSD fluctuations are likely due to the inherent flexibility of loop regions in RIPK1. We have included a more detailed analysis of RMSD and radius of gyration (Rg) in Supplementary Fig. 7 and revised the Results section accordingly.

Fig.S7 Molecular dynamics simulation stability metrics. Time-dependent root mean square deviation (RMSD) and radius of gyration (Rg) profiles for PB-AIM2^{HIN}, PB-AIM2^{PYD}, PB-ZBP1, and PB-RIPK1 simulated systems.

Comment 4: The efficiency of PANoptosis inhibition by PB@PM was similar to the other three specific programmed cell death inhibitors. What is the advantage of using PB@PM over these inhibitors? Please clarify the unique therapeutic value of PB@PM in PANoptosis modulation.

Response: We sincerely thank the reviewer for raising this critical point regarding the comparative advantage of PB@PM. While our data indeed show that the efficiency of PANoptosis inhibition by PB@PM is comparable to the combination of three specific programmed cell death inhibitors, PB@PM offers several distinct and translationally significant advantages that underscore its unique therapeutic value.

First, the core therapeutic superiority of PB@PM lies in its ability to concurrently and comprehensively suppress pyroptosis, apoptosis, and necroptosis within a single agent. As highlighted by the reviewer and supported by emerging evidence, extensive molecular crosstalk exists among these pathways. Monotherapeutic inhibition of one pathway often fails due to compensatory activation of alternative death mechanisms, thereby limiting overall efficacy[1–6]. Our in vitro data robustly demonstrate that under

equimolar low-dose conditions, the triple combination of Necrostatin-1 (necroptosis inhibitor), Disulfiram (pyroptosis inhibitor), and Z-VAD-FMK (pan-caspase inhibitor) only partially restored cell viability. In contrast, PB monotherapy achieved efficacy comparable to the triple combination and significantly outperformed any individual inhibitor (Supplementary Fig. 21D). This confirms that PB@PM effectively disrupts the compensatory crosstalk among distinct cell death pathways, enabling broad-spectrum PANoptosis inhibition via a single therapeutic entity—a feat difficult to achieve with a cocktail of small molecules due to differential pharmacokinetics and potential off-target effects.

Beyond this fundamental multimodal anti-PANoptotic activity, PB@PM provides critical pharmacological and practical benefits over conventional small-molecule inhibitors:

- (1) **Targeted Delivery and Enhanced Bioavailability:** The platelet membrane (PM) coating enables selective accumulation of PB@PM in the ischemic myocardium (Fig. 4G–I), thereby enhancing local drug concentration at the disease site, improving therapeutic efficacy, and minimizing potential off-target effects in healthy tissues.
- (2) **Superior Pharmacokinetics:** The biomimetic nature of the PM coating reduces rapid clearance by the mononuclear phagocyte system, prolongs systemic circulation, and improves the half-life of the nanoparticle[7–9]. In contrast, small-molecule inhibitors often suffer from short half-lives, rapid metabolism, and dose-limiting toxicities, which can hinder their clinical translation.
- (3) **Favorable Biosafety Profile:** Comprehensive assessments confirmed that PB@PM exhibits no significant systemic toxicity or histopathological alterations (Supplementary Fig. 11 and Supplementary Table 2), supporting its potential for safe clinical application.

Moreover, molecular dynamics simulations provide a structural rationale for the multi-target action of PB, revealing direct and high-affinity interactions with multiple core PANoptosome components (RIPK1, ZBP1, and AIM2) (**Fig. 3**). This multi-target engagement underpins its ability to simultaneously inhibit multiple PANoptosomes, a

mechanism distinct from the single-target action of specific small-molecule inhibitors.

In summary, PB@PM represents a targeted, multifunctional nanotherapeutic strategy that not only simultaneously inhibits PANoptosis crosstalk but also addresses key translational challenges—such as targeted delivery, sustained release, and reduced toxicity—associated with combination small-molecule therapies. Its design offers a unified solution to achieve synergistic PANoptosis inhibition, avoiding the complexities of dosing, scheduling, and potential drug interactions inherent in multi-drug regimens.

References

1. Sun, X. et al. PANoptosis: Mechanisms, biology, and role in disease. *Immunol. Rev.* 321, 246-262 (2024).
2. Sundaram, B. et al. NLRP12-PANoptosome activates PANoptosis and pathology in response to heme and PAMPs. *Cell* 186, 2783-2801.e20 (2023).
3. Sundaram, B. et al. NLRC5 senses NAD plus plus depletion, forming a PANoptosome and driving PANoptosis and inflammation. *Cell* 187, 4061-4077.e17 (2024).
4. Wang, Y. & Kanneganti, T. D. From pyroptosis, apoptosis and necroptosis to PANoptosis: A mechanistic compendium of programmed cell death pathways. *Comput. Struct. Biotechnol. J.* 19, 4641-4657 (2021).
5. Shi, C. X. et al. PANoptosis: A Cell Death Characterized by Pyroptosis, Apoptosis, and Necroptosis. *J. Inflamm. Res.* 16, 1523-1532 (2023).
6. Kuriakose, T. et al. ZBP1/DAI is an innate sensor of influenza virus triggering the NLRP3 inflammasome and programmed cell death pathways. *Sci. Immunol.* 1, aag2045 (2016).
7. E. Blanco, H. Shen, M. Ferrari, Principles of nanoparticle design for overcoming biological barriers to drug delivery. *Nature Biotechnology* 33, 941-951 (2015).
8. H. Yan et al., Engineering Cell Membrane-Based Nanotherapeutics to Target Inflammation. *Advanced Science* 6, (2019).
9. Li B. et al. A Nanocapsule System Combats Aging by Inhibiting Age-Related Angiogenesis Deficiency and Glucolipid Metabolism Disorders. *ACS Nano.* 2024;18(32):21061-21076.

Comment 5: How long does PB@PM persist in cardiac tissue? Additionally, please explain the primary mechanism by which PB@PM ameliorates fibrosis at the 28-day timepoint (e.g., modulation of inflammatory pathways, suppression of PANoptosis-related cytokines, or direct effects on fibroblast activation)

Response: We thank the reviewer for these insightful questions regarding the pharmacokinetics and anti-fibrotic mechanism of PB@PM.

Cardiac Retention of PB@PM: Our extended pharmacokinetic studies using Cy5.5-labeled PB@PM demonstrated robust nanoparticle accumulation in the ischemic myocardium for up to 5 days post-injection, with signals detectable even at later time points (Response Fig. R1). This prolonged retention is attributed to the active targeting mediated by platelet membrane proteins (e.g., CD41, CD42b, CD47) and the enhanced permeability and retention (EPR) effect in the inflamed cardiac microenvironment, ensuring sustained therapeutic presence at the injury site.

Mechanism of Anti-Fibrotic Action at Day 28: Our integrated analyses support a multi-faceted mechanism whereby PB@PM ameliorates fibrosis primarily through upstream modulation of cardiomyocyte death and inflammation, rather than direct effects on fibroblasts:

(1) Primary Driver: PANoptosis Inhibition and DAMP Reduction: The cornerstone of the anti-fibrotic effect is the significant reduction in cardiomyocyte PANoptosis by PB@PM (evidenced by decreased TUNEL⁺ cells in Fig. 5I-J and suppression of PANoptosome activation in Fig. 6J and Supplementary Figs. 18-19). By curtailing this major source of cell death, PB@PM fundamentally reduces the release of pro-fibrotic damage-associated molecular patterns (DAMPs) and intracellular contents that initiate and perpetuate the fibrotic cascade.

(2) Immunomodulation and Attenuation of Profibrotic Signaling: PB@PM potently attenuated the early inflammatory response by reducing the infiltration of pro-inflammatory M1 macrophages (CD86⁺) and neutrophils (Ly6G⁺) (Fig. 5K-N), and downregulating the TLR4/NF- κ B signaling axis (Supplementary Fig. 13). This shifts the cytokine milieu from a pro-fibrotic (high TNF- α , IL-1 β , IL-6) to a pro-reparative

(elevated IL-10, TGF- β) profile, thereby creating an environment less conducive to pathological fibroblast activation.

(3) Promotion of Reparative Fibrosis and Angiogenesis: Consequently, by mitigating excessive cell death and inflammation, PB@PM facilitated a more organized reparative process. This was evidenced by enhanced deposition of structural collagens (I and III), promotion of α -SMA⁺ myofibroblasts (key for wound contraction), and stimulation of CD31⁺ angiogenesis in the peri-infarct region (Fig. 5O-R, Supplementary Figs. 14-15). This coordinated response supports the formation of stable scar tissue, preserves ventricular integrity, and attenuates adverse remodeling.

While our data strongly support this cascade originating from cardiomyocyte protection, we acknowledge the reviewer's point and agree that potential direct effects of PB@PM on cardiac fibroblast activation represent an interesting avenue for future investigation.

Fig. R1 Typical ex vivo fluorescence images and quantitative analysis of the fluorescence intensities of damaged hearts at day 1, day 3, day 5, day 7, day 14, and day 28 after administration of Cy5.5-labeled PB@PM (n = 3). Data are expressed as mean \pm SD. P values are calculated using one-way ANOVA with Tukey correction. *P < 0.05, ***P < 0.001, and ns means no significance.

Comment 6: Mitochondrial damage plays a central role in PANoptosis. The authors claimed that PB@PM alleviates mitochondrial damage and inflammation, but did not explore the core mechanism of PB@PM ameliorating PANoptosis. Does PB@PM directly target PANoptosis or act through regulating mitochondrial dysfunction to suppress PANoptosis?

Response: We appreciate the reviewer for highlighting this central mechanistic question. Our results unequivocally demonstrate that PB@PM operates through a dual mechanism, involving both direct PANoptosome engagement and indirect protection via mitochondrial stabilization, which are likely synergistic.

1. Direct PANoptosome Targeting (The "Direct Hit"): (1) **Computational Evidence:**

Molecular dynamics simulations revealed that PB stably binds key functional domains of core PANoptosome components: the HIN domain of AIM2 (critical for dsDNA sensing), the Z α domain of ZBP1 (essential for nucleic acid binding), and the kinase domain of RIPK1 (Fig. 3). (2) **Biophysical Validation:** Surface plasmon resonance (SPR) confirmed high-affinity binding between PB and ZBP1 ($K_D = 3.04 \times 10^{-9}$ M) and RIPK1 ($K_D = 1.15 \times 10^{-8}$ M) (Fig. R2). (3) **Cellular Validation:** Lentiviral overexpression assays showed that PB colocalized with and suppressed the activation of AIM2, ZBP1, and phosphorylated RIPK1 under both normoxic and OGD/R conditions (Supplementary Fig. 20). Furthermore, PB@PM treatment significantly inhibited the colocalization of ASC with caspase-1/8, a key step in PANoptosome assembly (Supplementary Fig. 21K-L). (4) **Functional Outcome:** Multiplex immunoblotting confirmed that PB@PM downregulated core components and downstream effectors across all three PANoptosomes (Fig. 6J, Supplementary Figs. 18–19).

2. Mitochondrial Protection and Metabolic Rescue (The "Upstream Blocker"): (1)

ROS Scavenging: Cardiomyocytes are rich in mitochondria, which are major sources of ROS following ischemia-reperfusion injury [1-3]. PB@PM exhibited potent reactive oxygen species (ROS)-scavenging activity, mimicking superoxide dismutase (SOD) and catalase (CAT) functions. It effectively eliminated \bullet OOH and \bullet OH radicals (Supplementary Fig. 26A–D) and reduced intracellular ROS levels in OGD/R-injured cardiomyocytes (Supplementary Fig. 26F–H). (2) **Mitochondrial Functional Restoration:** PB@PM restored mitochondrial membrane potential (Supplementary Fig. 27A-C) and enhanced mitochondrial oxidative phosphorylation, as demonstrated by increased basal and maximal oxygen consumption rate (OCR) and ATP production (Supplementary Fig. 27D-G). This metabolic rescue is crucial since mitochondrial

dysfunction and mtDNA release are key upstream triggers for PANoptosome activation (e.g., via the AIM2 and ZBP1 sensors).

Thus, PB@PM directly disrupts PANoptosome assembly via multi-target molecular interactions, while concurrently preserving mitochondrial integrity and reducing oxidative stress. This dual action synergistically attenuates cardiomyocyte death and disrupts the inflammatory amplification loop that fuels PANoptosis. We have clarified this integrative mechanism in the revised manuscript.

Fig. R2 Surface plasmon resonance (SPR) analysis showing the binding affinity of PB to ZBP1 (A) and RIPK1 (B).

Fig.S26 PB@PM scavenges ROS. (A-C) Electron spin resonance (ESR) detection of hydroperoxyl ($\bullet\text{OOH}$) and hydroxyl ($\bullet\text{OH}$) radicals. (D) Visual observation of PB reactivity with H_2O and H_2O_2 . (E) Flow cytometry analysis of phagocytic efficiency of

FITC-labeled PB@PM in HL-1 cells. (F-H) Fluorescence imaging (F; Scale bar: 50 μm) and flow cytometry of ROS levels in OGD/R-injured HL-1 cells.

Fig.S27 PB@PM restores mitochondrial function. (A-C) JC-1 staining images (Scale bar: 50 μm) and flow cytometry analysis of mitochondrial membrane potential (MMP). (D-G) Oxygen consumption rate (OCR) quantification in PB@PM-treated HL-1 cells (n = 5-10 biologically independent replicates). Data: mean \pm SD. Significance: one-way ANOVA with Tukey correction (*P < 0.05, **P < 0.01, ***P < 0.001).

References

- Chen Q. et al. Nrf3-Mediated Mitochondrial Superoxide Promotes Cardiomyocyte Apoptosis and Impairs Cardiac Functions by Suppressing Pitx2. *Circulation*. 2025;151(14):1024-1046.
- Ye T. et al. Protective effects of Pt-N-C single-atom nanozymes against myocardial ischemia-reperfusion injury. *Nat Commun*. 2024;15(1):1682.
- Zheng Y. et al. Mitochondria-Targeted ROS Scavenging Natural Enzyme Cascade Nanogels for Periodontitis Treatment via Hypoxia Alleviation and Immunomodulation.

Comment 7: What is the underlying mechanism of PB@PM reprogramming mitochondria metabolism? Does it directly bind with mitochondrial proteins or modulate mtDNA release?

Response: We thank the reviewer for raising this important mechanistic question. Cardiomyocytes are indeed highly enriched in mitochondria, which serve as the primary source of reactive oxygen species (ROS) following ischemia-reperfusion injury. Our data demonstrate that PB nanoparticles possess multi-enzyme mimetic activities—including superoxide dismutase (SOD), catalase (CAT), and hydroxyl radical scavenging—as confirmed by electron spin resonance (ESR) spectroscopy (Supplementary Fig. 26A–D). Consequently, PB@PM effectively quenched intracellular ROS in OGD/R-injured HL-1 cardiomyocytes (Supplementary Fig. 26F–H). Functionally, PB@PM treatment restored mitochondrial membrane potential (Supplementary Fig. 27A–C) and enhanced mitochondrial respiratory capacity and ATP production, as evidenced by Seahorse XF analysis (Supplementary Fig. 27D–G), collectively indicating a significant improvement in mitochondrial metabolic efficiency. Currently, we have no experimental evidence indicating that PB@PM directly binds mitochondrial proteins or modulates mtDNA release. Myocardial mitochondrial metabolic reprogramming involves a complex network of pathways, including glycolysis, the TCA cycle, and fatty acid β -oxidation, which warrant further systematic investigation. We fully agree with the reviewer that elucidating the precise molecular interplay between PB nanoparticles and mitochondrial components—particularly regarding mtDNA release and its role in PANoptosis activation—represents an important direction for future research.

Comment 8: Whether PB@PM truly block PANoptosome assembly or not should be further verified. Current data only showed inhibition of downstream cell death markers, but failed to demonstrate the disruption of PANoptosome complex.

Response: We appreciate the reviewer's insightful comment regarding the need for

direct evidence of PANoptosome disruption. To address this, we performed additional immunofluorescence analyses to assess the colocalization of ASC with caspase-1 and caspase-8—key events in PANoptosome assembly. Our results clearly show that both PB and PB@PM treatment significantly reduced the colocalization of ASC with caspase-1/8 in OGD/R-stimulated HL-1 cells (Supplementary Fig. 21K,L), providing direct visual evidence that PB@PM inhibits PANoptosome complex formation. Moreover, our molecular dynamics simulations revealed high-affinity binding between PB and core PANoptosome components—RIPK1, ZBP1, and AIM2 (Fig. 3). Consistent with this, western blot analyses demonstrated that PB@PM downregulated the active forms of these proteins and their downstream effectors (e.g., cleaved caspases, phosphorylated RIPK1/RIPK3/MLKL, and GSDMD) in both *in vivo* and *in vitro* models (Fig. 6J, Supplementary Figs. 18–21). Together, these multimodal data support the conclusion that PB@PM interferes with PANoptosome assembly. We acknowledge that further dynamic or structural studies would be valuable, and we plan to incorporate such approaches in future work as relevant technologies advance.

Fig.S21 PB@PM inhibits PANoptosis *in vitro*. (K-L) Confocal microscopy images showing the colocalization of ASC and caspase-1/8 in OGD-R-induced HL-1 cells following PB or PB@PM treatment. (n = 3 biologically independent replicates).

Comment 9: Molecular dynamics simulations suggested that PB binds with RIPK1, ZBP1 and AIM2, while the direct evidence of PB disrupting PANoptosome complex assembly was absent.

Response: We thank the reviewer for emphasizing the need for direct evidence linking PB binding to PANoptosome disruption. As noted in our response to Comment 8, we have now provided confocal microscopy data showing that PB@PM significantly suppresses the colocalization of ASC with caspase-1/8 (Supplementary Fig. 21K,L), a key step in PANoptosome assembly. This result offers direct cellular evidence that PB@PM inhibits the formation of functional PANoptosome complexes.

In addition, molecular dynamics simulations confirmed stable binding interactions between PB and the core PANoptosome components RIPK1, ZBP1, and AIM2 (Fig. 3). Subsequent biochemical validations showed that PB@PM treatment consistently downregulated the activation of downstream effectors across all three PANoptosome types and reduced levels of critical adaptors such as ASC and FADD (Fig. 6J, Supplementary Figs. 18–21). Collectively, these findings support the model that PB nanoparticles disrupt PANoptosome assembly via direct binding to key components.

Comment 10: The observed reduction in TNF- α /IL-1 β by PB@PM derives from PANoptosis inhibition or direct immune suppression should be further elucidated.

Response: We appreciate the opportunity to clarify the origin of the anti-inflammatory effects of PB@PM. Our *in vitro* experiments confirmed that PB@PM directly suppresses proinflammatory cytokine secretion (TNF- α , IL-1 β , IL-6) in LPS-stimulated RAW264.7 macrophages (Supplementary Fig. 12), consistent with prior studies [1-4]. To clarify whether PANoptosis inhibition contributes to reduced inflammation, we conducted a co-culture experiment where OGD/R-injured HL-1 cardiomyocytes (upper chamber) treated with PB@PM led to significant decreases in IL-1 β , IL-6, and TNF- α in RAW264.7 macrophages (lower chamber) (Supplementary Fig. 22). Since the macrophages were not directly exposed to PB@PM in this setup, the observed cytokine reduction can be attributed to the suppression of PANoptosis in cardiomyocytes, which

in turn attenuates immunogenic cell death and subsequent inflammation. Thus, the anti-inflammatory effect of PB@PM is multimodal, involving both direct cytokine suppression and inhibition of PANoptosis-mediated immunogenic cell death.

Fig.S12 PB@PM modulates inflammatory cytokine secretion. Enzyme-linked immunosorbent assay (ELISA) quantification of IL-1 β , IL-6, TNF- α , IL-10, and TGF- β levels in lipopolysaccharide (LPS)-stimulated RAW264.7 model treated with PB@PM (n = 3–4 biologically independent replicates). Data: mean \pm SD. Significance: one-way ANOVA with Tukey’s post-hoc test (*P < 0.05, **P < 0.01, ***P < 0.001).

Fig.S22 PB@PM can decrease inflammatory cytokine secretion via inhibiting PANoptosis. ELISA quantification of IL-1 β , IL-6, and TNF- α levels in RAW264.7 cells (in the lower layer) following treatment of OGD-R stimulated HL-1 cardiomyocytes (in the upper layer) using PB or PB@PM (n = 3 biologically independent replicates). Data: mean \pm SD. Significance: one-way ANOVA with Tukey’s post-hoc test (***P <

0.001; ns, $P > 0.05$).

References

1. Hou, R. et al. Prussian Blue Nanozyme Promotes the Survival Rate of Skin Flaps by Maintaining a Normal Microenvironment. *ACS Nano* 16, 9559-9571 (2022).
2. Zhang, K. et al. Hollow Prussian Blue Nanozymes Drive Neuroprotection against Ischemic Stroke via Attenuating Oxidative Stress, Counteracting Inflammation, and Suppressing Cell Apoptosis. *Nano Lett.* 19, 2812-2823 (2019).
3. Wang, F. et al. Polymer-modified DNA hydrogels for living mitochondria and nanozyme delivery in the treatment of rheumatoid arthritis. *Bioact Mater.* 47, 448-459 (2025).
4. He, H. et al. Design of a Multifunctional Nanozyme for Resolving the Proinflammatory Plaque Microenvironment and Attenuating Atherosclerosis. *ACS Nano.* 2023; 17(15): 14555-14571.